behaviour/cognition/psychology

animal cognition, research site, replicability, selection line, spatial cognition

**Authors for correspondence:**
K. Rosenberger
e-mail: katrinarosenberger91@gmail.com
C. Nawroth
e-mail: nawroth.christian@gmail.com

# Performance of goats in a detour and a problem-solving test following long-term cognitive test exposure

K. Rosenberger[1,2], M. Simmler[3], J. Langbein[4], N. Keil[1] and C. Nawroth[4]

[1]Swiss Federal Veterinary Office, Centre for Proper Housing of Ruminants and Pigs, Agroscope, 8355 Ettenhausen, Switzerland
[2]Graduate School for Cellular and Biomedical Sciences, University of Bern, 3012 Bern, Switzerland
[3]Digital Production Group, Agroscope, 8355 Ettenhausen, Switzerland
[4]Research Institute for Farm Animal Biology, Institute of Behavioural Physiology, 18196 Dummerstorf, Germany

KR, 0000-0003-4053-187X; MS, 0000-0002-4095-4111; JL, 0000-0002-1170-5431; NK, 0000-0002-3663-7682; CN, 0000-0003-4582-4057

Cognitive research in long-lived species commonly involves using the same animals in different experiments. It is unclear whether the participation in cognitive tests can notably alter the performance of individuals in subsequent conceptually different tests. We therefore investigated whether exposure to cognitive tests affects future test performance of goats. We used three treatment groups: goats with long-term exposure to human-presented object-choice tests (for visual discrimination and reversal learning tests + cognitive test battery), goats that were isolated as for the test exposure but received a reward from the experimenter without being administered the object-choice tests, and goats that were isolated but neither received a reward nor were administered the tests. All treatment groups were subsequently tested in two conceptually different cognitive tests, namely a spatial A-not-B detour test and an instrumental problem-solving test. We tested dairy goats, selected for high productivity, and dwarf goats, not selected for production traits, each at the same two research sites. We did not find notable differences between treatments with respect to the goats' detour or problem-solving performance. However, high variation was observed between the research sites, the selection lines, and among individuals, highlighting potential pitfalls of making accurate comparisons of cognitive test performances.

# 1. Introduction

Comparative approaches that identify key socio-ecological drivers of certain cognitive traits are important to understand the evolutionary origins of human cognition. Comparative research involving cognitive tests in non-human species is targeting animals in research facilities, including laboratory, zoo, and farm animals [1]. Depending on the species, this line of research commonly uses the same animals repeatedly over several experiments, which is often a necessity due to restricted financial resources and/or constraints in the available number of animals at the facility. Although it is known that animals can pick up learned contingencies over similar tests [2,3], it is still unclear whether participation in cognitive tests alters the performance of individuals in subsequent conceptually different cognitive tests. This raises the question whether potential long-term behavioural change due to repeated cognitive testing hampers comparability of data and replicability of study findings that are obtained from subjects with different histories of test exposure.

To answer this question, a variety of potentially confounding factors that can affect test performance in subsequent cognitive tests need to be disentangled. First, the cognitive testing *per se* may have an effect on future test performance. Frequent operation of similar cognitive tests enhances the ability to learn the test-inherent contingencies and thus alter test performances when compared with naive subjects: the recall of previously learned information can help to facilitate subsequent learning in a conceptually similar following test [2,3]. For instance, prior experience in detour tests with transparent obstacles also improved performance of pheasants in a subsequent novel detour test [4]. Although this effect of 'learning to learn' was shown for similar tests, little is known on how test exposure affects future performance in cognitive tests of a different type.

Research suggests that experience acquired via one specific test is probably domain-specific and barely transferable to different tests [5–7]. For instance, tool-use training in monkeys did improve performance in the physical cognition domain, but not their ability to generalize this tool-use knowledge to a novel tool [6]. Similar results were found in humans where participants slightly improved their performance in a cognitive test most similar to the test learned, but general transfer from one test to a conceptually different test was poor [5]. However, it is possible that the experience of short testing periods over one to two months in the previously mentioned studies [5–7] was not sufficient to induce biologically meaningful differences in test performance in subsequent tests. For example, highly trained dogs, such as competitive-level sport dogs or certified working dogs, were shown to be more persistent and successful in solving a problem than non-trained dogs [8–10]. However, other factors associated with the dog's environment such as rearing history might have caused changes beyond test experience.

Furthermore, cognitive testing is linked to more general changes in behaviour and physiology that could indirectly affect the performance in subsequent cognitive tests. Langbein *et al*. [11] found that offering cognitive challenges via a computer-based learning device to goats induced changes in vagal activity of the heart during successful learning, suggesting that operating the learning device was experienced as positive stress by those animals. The use of a 'call feeding station' to cognitively challenge pigs over a 12-week period not only led to an increase in locomotor behaviour and a decrease in belly nosing in the pigs' home environment, but also altered behaviour in subsequent open-field and novel-object tests. The cognitively challenged pigs showed reduced activity and excitement in these tests compared with control pigs that were not administered the cognitive challenge [12]. The effect was more pronounced after 12 weeks of access to the call feeding station than after six weeks. Previous experiences with cognitive tests might, therefore, alter the motivation to participate and, subsequently, the performance in future cognitive tests by reducing neophobia and/or stress levels.

In addition to previous experiences with cognitive testing itself, habituation to humans and to isolation in a test environment are important non-cognitive factors that can potentially cause relevant differences in test performance between habituated and naive individuals, because they also affect the motivation to participate in tests and the stress level in the test situation. By contrast to studies using an automated reward delivery [11–13], other cognitive test paradigms, such as object-choice tests, often require human–animal interaction, for example for positive reinforcement by a human experimenter. This interaction may be stressful for animals if they have not been habituated or if they had negative experiences with humans in the past. For example, high emotional reactivity towards the experimenter was found to affect the learning performance of pigs [14]. In addition, separation from the group is stressful for most social animals and has been found to increase vocalizations, heart rate, and cortisol levels [15–19]. A lack of sufficient habituation to humans and isolation can thus hamper a correct and reliable assessment of cognitive performance by a decrease in attention and participation [20,21].

Goats are a promising model species for cognitive research. By means of object-choice tests, they have been found to use direct and indirect information to locate hidden food rewards and anticipate the trajectory of hidden objects [22,23]. They have been shown to be good problem-solvers in visual discrimination and spatial detour tests [11,24–26], but also in instrumental problem-solving tests (PSTs) that involved opening a box or container [27,28]. In addition, domestic goats have been shown to interact with humans in complex ways [29]. Thus, object-choice tests as well as detour and PSTs have been shown to be suitable tests to assess cognitive capacities in goats.

In the current study, we investigated whether long-term exposure to object-choice tests affects the performance of goats in subsequent conceptually different cognitive tests. To control for habituation to humans and isolation in a test environment, we conditioned three treatment groups: goats with long-term exposure to human-presented object-choice tests (COG treatment), goats that were isolated as for the test exposure but received a reward from the experimenter without being administered the object-choice tests (POS treatment), and goats that were isolated but neither received a reward nor were administered the tests (ISO treatment). All treatment groups were subsequently tested in two conceptually different cognitive tests, namely a spatial A-not-B detour test (ABT) and an instrumental PST. The ABT requires animals to learn to detour around a spatial barrier before (=A trials) and after (=B trials) the position of the barrier is altered [30,31]. The PST in our study is an instrumental manipulation test that requires the animal to open a familiar food container covered with a lid novel to the animal. To increase the heterogeneity of our sample and thus the external validity of our findings, we tested each of two different selection lines of goats (dairy goats and dwarf goats) at the same two research sites [32,33].

We hypothesized that previous cognitive test exposure improves performance in an ABT and a PST, which measure behavioural flexibility and problem-solving abilities, respectively. We thus expected the COG group to perform better than the POS group in the tests. We furthermore hypothesized that positive human–animal interaction improves performance in these tests. Correspondingly, we expected the POS group to outperform the ISO group.

# 2. Material and methods

## 2.1. Location, animals and housing conditions

To increase external validity [32–34], the study was carried out at two research sites, at the Centre for Proper Housing of Ruminants and Pigs at Agroscope in Ettenhausen (ET), Switzerland, and at the Research Institute for Farm Animal Biology in Dummerstorf (DU), Germany, and with two selection lines of goats, namely Nigerian dwarf goats (61 non-lactating female goats) and dairy goats (59 non-lactating female goats). The Nigerian dwarf goat is commonly kept as a pet and zoo animal in Europe and not selected for productivity traits. We used dwarf goats bred at the research institute in DU. The only selection aim in this population was to avoid inbreeding. The potential milk yield of dwarf goats probably does not exceed 0.3 kg per day [35]. As it was common practice in DU, dwarf goat kids stayed with their dams for six weeks before they were weaned. Additionally, we used three of the most common high-producing dairy breeds in Switzerland and Germany, and their cross-breeds, namely Saanen ($n = 15$), Chamois Coloured ($n = 12$), Saanen × Chamois ($n = 3$), and Deutsche Edelziege ($n = 29$). These animals had a potential milk yield of up to 3 kg per day [36]. In accordance with common practice in the dairy goat industry, the dairy goat kids had been separated from their dam shortly after birth and were artificially raised.

At the Agroscope research station in ET, we housed 30 dwarf goats and 30 dairy goats (15 Saanen, 12 Chamois Coloured, 3 Saanen × Chamois cross-breeds, see electronic supplementary material, file T1_animals). The dwarf goats were born between January and February 2017 in DU, Germany, and moved to ET in June 2017. The dairy goats were born between February and April 2017 on different Swiss farms and were moved to ET in June/July 2017. At the location in DU, we housed 31 dwarf goats (Nigerian dwarf) and 29 dairy goats (Deutsche Edelziege). The dwarf goats were born between January and March 2018 in DU, except for eight animals. These were bought from the Zoo Osnabrück and the Wildpark Lüneburger Heide, Germany, due to a shortage of female animals in the facility's own breeding stock. All dairy goats were born on the same German farm in March 2018 (Gleistal-Mutterkuhhaltungs GmbH, Golmsdorf) and were moved to DU in July 2018.

All goats were moved to pens of 9–11 goats at the age of seven to eight months: three pens of dairy goats and three pens of dwarf goats at each research site. The total area of each dwarf goat pen was 14 m² (approx. 3.6 × 3.9 m), consisting of a deep-bedded straw area of 11 m² (approx. 2.8 m × 3.9 m) and a 0.5 m elevated feeding place (1.4 m²). The total area of each dairy goat pen was 17.7 m² (approx. 3.9 m × 4.55 m)

consisting of a deep-bedded straw area of 13.4 m$^2$ (approx. 4.55 × 2.95 m) and a 0.65 m elevated feeding place (1.82 m$^2$). Hay was provided behind a feeding fence at the feeding place twice a day at around 8.00 and 16.00 in ET and at around 7.00 and 13.00 in DU. Each pen had one watering place and a mineral supply. Additional structures in the straw-bedded area included a wooden bench (for dairy: 2.4 m long, 0.6 m high, 0.62 m wide; for dwarf: 2.3 m long, 0.5 m high, 0.5 m wide) along the wall of the pen and a round wooden table (0.8 m high, 1.1 m in diameter) in the centre of the pen.

All animal care and experimental procedures were performed in accordance with the relevant Swiss legislative and regulatory requirements as well as the German welfare requirements for farm animals and the ASAB/ABS Guidelines for the Use of Animals in Research [37]. All procedures involving animal handling and treatment were approved by the Cantonal Veterinary Office, Thurgau, Switzerland (Approval no. TG04/17–29343), and the Committee for Animal Use and Care of the Ministry of Agriculture, Environment, and Consumer Protection of the federal state of Mecklenburg-Vorpommern, Germany (Approval no. 7221.3-1.1-062/17).

## 2.2. Treatment groups and procedures

Three goats from each of the 12 pens were pseudo-randomly assigned to one of the three treatment groups: COG ($n = 36$), POS ($n = 36$), and ISO ($n = 36$). Except one pen, all pens housed one to two extra goats not assigned to a treatment group to replace others in case of e.g. disease or injury. In 44 test sessions, distributed over a period of four to five months, the COG group was exposed to cognitive tests in the form of discrimination and reversal learning tests and a cognitive test battery (see electronic supplementary material, table S1 for more details). During these tests, COG goats received food rewards from the experimenter for correct responses. The POS group was not exposed to cognitive tests but received a similar number of rewards as the individuals in the COG group (=median number of rewards received by COG group in the previous test session), provided by the experimenter in the test arena at pseudo-random times and over a similar period of being isolated as the COG group (=median time taken by COG group to finish all trials in the previous test session). Contrasting COG versus POS allows investigating the effect of the cognitive testing itself, disentangled from the effects of the positive association with the human and the isolation from the group during testing. Individuals allocated to the ISO treatment neither participated in cognitive tests nor did they receive rewards by the experimenter. However, they were isolated over a similar period as the COG and the POS group in the same arena (=median time taken by COG group to finish all trials in the previous test session) and with the experimenter present behind the crate, as was the case for the COG and POS treatments. Contrasting POS versus ISO allows investigating the effect of the positive association with the human, disentangled from the effect of isolation from the group during testing. To control for caloric intake, ISO animals received a similar amount of food as the COG and POS goats, but to avoid positive association with the human, the food was provided scattered over the floor of the waiting room where goats were kept before they were individually isolated in the test arena.

All goats in ET were between 15 and 18 months old when tested in the ABT (mean ± s.d.: dairy goats: 494 ± 3 days, dwarf goats: 537 ± 1 days) and in the PST (mean ± s.d.: dairy goats: 491 ± 3 days, dwarf goats: 533 ± 1 days). In DU, the goats were between 19 and 20 months old in the ABT (mean ± s.d.: dairy goats: 586 ± 0 days, dwarf goats: 616 ± 3 days) and in the PST (mean ± s.d.: dairy goats: 579 ± 0 days, dwarf goats: 608 ± 3 days).

## 2.3. A-not-B detour test

The ABT according to Osthaus *et al.* [30] is a test that requires animals to learn to detour around a spatial barrier before (=A trials) and after (=B trials) the placement of the barrier is altered. Thus, the animal is required to suppress the once successful response (in A trials) and adapt to the new spatial set-up (B trials). Due to the nature of the ABT, the animals are expected to show a spatial perseveration error in the B trials following a variable number of A trials with the magnitude of the error depending on their level of impulse control. It is assumed that animals that more strongly suppress their previously learned response make fewer errors.

### 2.3.1. Test set-up and procedure

The experiment was conducted in a large rectangular arena, which was divided by a movable fence into two same-sized compartments while leaving a gap on one or the other side of the arena (figure 1). The

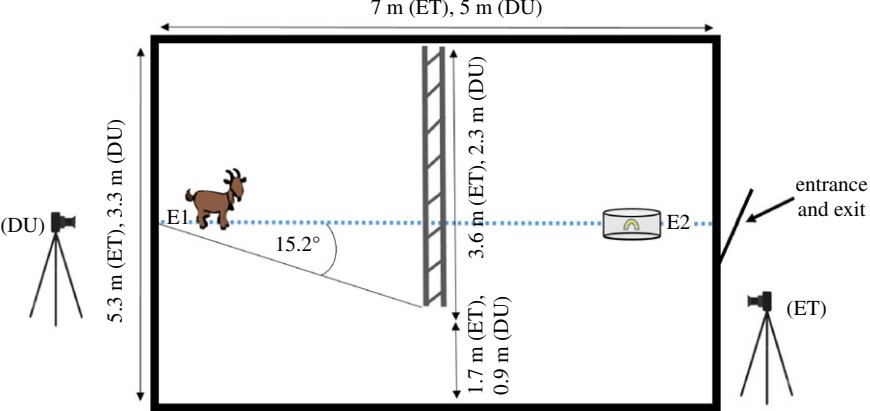

**Figure 1.** Schematic drawing of the test arena used for the A-not-B detour test in Ettenhausen (ET) and Dummerstorf (DU). Positions of the two experimenters are indicated with E1 and E2. The blue dotted line indicates the imaginary line that was used to determine whether a goat stepped towards the correct or the incorrect direction (=Accuracy).

size of the arena differed slightly in ET and DU, but the angle between test subject and gap location was kept the same (approx. 15.2°) by adjusting the fence length accordingly (figure 1). In ET, a mobile pen, familiar to the goats, was used to move the tested individuals in groups of 4–10 to the test arena. The mobile pen served as a holding area next to the test pen while individual goats were being tested. In DU, the test arena was much closer to the home pens of the goats, hence the goats were led individually to the test arena. In the arena, the goat was taken to the opposite side of the arena and restrained by experimenter 1 (E1; figure 1 and electronic supplementary material, figure S1). A second experimenter (E2) stayed at the side of the entrance, shaking a container, familiar to the goat, with dry pasta to motivate the goat to walk through the gap towards the food. The trial started when E1 released the goat. Each goat received eight trials: during the so-called 'acquisition phase', the goat received four trials with the gap on one side of the arena (=Trials A1–A4). Subsequently, the fence was moved, and the goat received four trials with the gap on the other side of the arena (=Trials B1– B4). If the tested goat moved through the gap, it was allowed to feed for approximately 5 s from the container before it was led back to the starting point through the same gap. Between the A and B trials, E2 switched the position of the gap, while E1 was holding the tested goat and covered its eyes. For half of the goats, the gap was first on the left side and for the other half it was first on the right side of the arena.

### 2.3.2. Behavioural measures

Two parameters of performance were determined: (i) *Accuracy*, a binary response being either correct (1) or incorrect (0) depending on whether the first step over an imaginary line (figure 1) was or was not directed towards the side of the arena where the gap was located and (ii) *Latency*, defined as the time (in seconds) from the first step until passing the gap in the fence with the shoulders. If a goat did not pass the gap in less than 60 s after it had been released, the trial was coded as *not available* and the animal was taken back to the starting point and the next trial began.

## 2.4. Problem-solving test

The set-up and procedure of the PST are less standardized than the ABT and vary considerably between different studies. Experimental PST set-ups differ for example in the type of device used, such as containers, puzzle boxes and tubes, as well as in the required opening techniques, including pulling a stick, sliding a lid, moving a lever and rotating a lid [38–41]. The PST test in our study is an instrumental manipulation test that requires the animal to open a familiar unfixed and freestanding food container covered with a lid novel to the animal.

### 2.4.1. Habituation phase

A round plastic container (11.5 cm tall, 34 cm in diameter) with a light wooden lid (38 cm in diameter, 0.33 kg) was used as manipulandum for the PST. The goats were already familiar with this container,

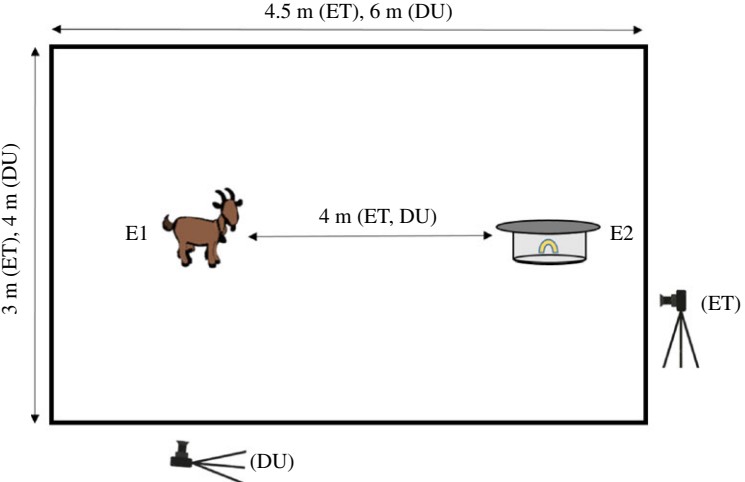

**Figure 2.** Schematic drawing of the test arena used for the problem-solving test in Ettenhausen (ET) and Dummerstorf (DU). Positions of the two experimenters are indicated with E1 and E2.

and the notion that it contains food, from their general handling routines. During the habituation phase, the container was open with the lid leaned against the container. The goats were allowed to investigate the container and the lid, first in pairs, whereby the container was baited 10 times with one piece of pasta per goat. Once they fed out of the container repeatedly, they were allowed to investigate it individually until they had reliably eaten the reward out of the container in five consecutive approaches.

### 2.4.2. Test set-up and procedure

For the test phase, the container was covered with the wooden lid such that physical manipulation was necessary to open the container and access the food reward. Before the start of the test phase, each goat was individually led into the test arena (figure 2) and allowed to eat a piece of pasta from the open container. One experimenter (E1) then restrained the goat on one side of the test arena, while another experimenter (E2) placed a piece of pasta into the container and covered it with the lid. Subsequently, the goat was released and allowed to approach the container. Each goat received five consecutive trials within a single day. A motivation trial with the lid open was administered between all test trials to maintain motivation.

### 2.4.3. Behavioural measures

As measures of motivation to engage with the test, we recorded (i) whether the goat touched the lid or not (=*Touched*; with any body part) and (ii) the latency to touch (=*LatencyT*) measured from release to first touch of the lid (with any body part). As measures of proficiency, we recorded (i) whether the goat opened the container or not (=*Opened*) and (ii) the latency to open (=*LatencyO*) defined as the time from first touch with any body part to opening the lid. If a goat did not approach the container within a 1 m radius within 30 s after it had been released, the trial was coded as *not available* and the animal was taken back to the start for the next trial.

## 2.5. Data coding and statistical analysis

All trials were videotaped with a camcorder (ET: Sony HDR-CX240E; DU: Panasonic HDC-SD60), and most parameters were additionally recorded live by the experimenters. For the analysis of the ABT, we used the live-recorded data (electronic supplementary material, file T2_ABT). For the PST, we decided in retrospect to code additional variables from video and, therefore, used the more comprehensive video-coded data for analysis (electronic supplementary material, file T3_PST).

To perform the reliability analysis for the ABT, an external person not familiar with the hypotheses coded 50% of the trials from videos. Inter-observer reliability was found to be very high (*Accuracy*: $\kappa = 0.966$, $z = 14.2$, $p < 0.001$; *Latency*: $r = 0.99$, $p < 0.001$). For the PST, only the variables *Opened* and *LatencyO* were coded live, thus we used these two measures to analyse inter-observer reliability and compared

them with the corresponding data coded by an external person from video. Again, inter-observer reliability was found to be very high (*Opened*: $\kappa = 0.903$, $z = 10.4$, $p < 0.001$; *LatencyO*: $r = 0.98$, $p < 0.001$).

All statistical analyses were performed in R v. 4.0.3 [42]. In the ABT and PST, seven dwarf goats and one dairy goat did not participate because they were too stressed during training for the preceding object-choice tests and thus could not receive their assigned treatment. In the ABT, we excluded nine animals because they jumped over or crawled under the fence, or due to latencies above 60 s. In addition, goats that showed an incorrect response in A3 and A4 trials were excluded from the analysis ($n = 19$) because they probably had not learnt the correct response by the end of the A trials and were thus, by definition, not able to show the perseveration error in the subsequent B trials. In the PST, we excluded 29 individuals during training before the actual PST test because they did not meet the training criteria (= reliably eating the reward out of the open container within five approaches) and four more animals that had missing trials due to human errors or technical failure. In sum, statistical analysis was performed on data of 28 COG, 22 POS and 27 ISO goats in the ABT and 23 COG, 24 POS and 20 ISO goats in the PST.

To test for the perseveration error in the ABT, we applied one-sided paired McNemar tests (*exact2x2* function from the R package exact2x2 [43]) and one-sided paired Wilcoxon signed-rank tests (base R function *wilcox.test*). McNemar tests were used to test whether B1 trials were less frequently correct than the corresponding A4 trials (binary variable *Accuracy*). Similarly, Wilcoxon signed-rank tests were used to test whether B1 trials were characterized by longer *Latency* as compared with the A4 trials.

To analyse the effects of the treatments (COG, POS, ISO) on the dependent variables in the ABT and the PST, we employed linear mixed-effects models from the R package lme4 [44]. For binary responses, i.e. *Accuracy* in the ABT and *Touched* and *Opened* in the PST, the models were estimated as generalized linear mixed model (GLMM) with logit link by using the *glmer* function. For the continuous responses, i.e. *Latency* to cross the fence in the ABT and *LatencyT* and *LatencyO* in the PST, the models were estimated as ordinary linear mixed model by using the *lmer* function. We visually inspected residuals of all models by using the package DHARMa [45]. To achieve better normal distribution of residuals, we $\log_2$-transformed the latencies before model fitting.

For the ABT model, formulae in lme4 syntax were as follows:

$$response \sim 0 + Type{:}Treatment + Type{:}Treatment{:}I(Trial - 1) + (1|SelectionLine)$$
$$+ (1|Site/Pen/Individual).$$

We included an intercept for each type of trial (=A and B trials) and treatment interaction individually (0 + Type:Treatment) and a slope for trial number for each type–treatment interaction (Type:Treatment:I[Trial − 1]). The trial number was included as Trial − 1 to render the intercept to correspond to Trial 1 instead of the non-meaningful Trial 0. Besides these fixed effects, a random intercept for selection line (1 | SelectionLine) and for individual nested within the pen and within site (1 | Site/Pen/Individual) was included to account for repeated testing and a potential effect of the affiliation to pen (A–F, U–Z) and site (ET, DU).

For the PST, the model formulae in lme4 syntax were as follows:

$$response \sim 0 + Treatment + Treatment{:}I(Trial - 1) + (1|SelectionLine) + (1|Site/Pen/Individual).$$

Here, we included an intercept for each treatment (0 + Treatment) and a slope for trial for each treatment (Treatment:I[Trial − 1]). Besides these fixed effects, a random intercept for individual nested within the pen and within site (1 | Site/Pen/Individual) was included to account for repeated testing and potential effects of pen and site affiliation. Only dairy goats opened the container and could, therefore, be analysed with respect to the corresponding behavioural responses. By contrast to all other models, the models for *Opened* and *LatencyO* have thus no random intercept for selection line. To investigate differences in all behavioural responses between the treatments, we tested treatment contrasts for the fixed effects with the *glht* function from the R package multcomp [46]. The *p*-values for fixed-effect estimates and for the contrasts were obtained by using Wald *z*-tests (*summary.ghlt* function, multcomp package).

# 3. Results

## 3.1. A-not-B detour test

In the ABT, we recorded *Accuracy* (figure 3) and *Latency* to cross the fence (figure 4). For both behavioural measures, the individual variability in B trials was large and no consistent patterns were apparent for

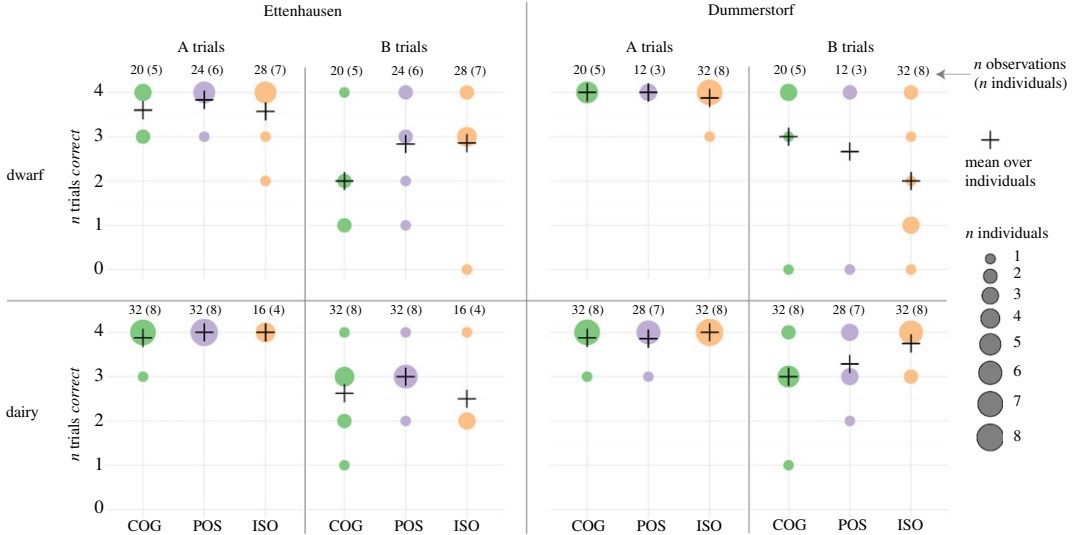

**Figure 3.** Accuracy shown as the number of correct trials for dwarf (top panels) and dairy goats (bottom panels) of the COG, POS and ISO treatment groups during A trials and B trials in Ettenhausen (four left panels) and Dummerstorf (four right panels). The size of circles indicates the number of animals. As in the statistical analysis, goats that showed an incorrect response in A3 and A4 trials are excluded from this figure.

treatment groups, selection lines and sites. A spatial perseveration error was indicated for all treatment groups: For B1 trials as compared with A4 trials, the McNemar test suggested a lower proportion of correct responses (COG: $p < 0.001$, POS: $p = 0.002$, ISO: $p < 0.001$) and the Wilcoxon signed-rank test a higher latency (COG: $p = 0.006$, POS: $p < 0.001$, ISO: $p = 0.009$).

According to the GLMM results, treatments did not differ in their *Accuracy*, neither in A (POS–COG: $p = 0.83$, ISO–COG: $p = 0.87$, ISO–POS: $p = 0.96$; electronic supplementary material, table S2) nor in B trials (POS–COG: $p = 0.13$, ISO–COG: $p = 0.19$, ISO–POS: $p = 0.81$; electronic supplementary material, table S2). Consistent across the three treatment groups, the probability for the goats to choose the correct side increased with increasing number of B trials, but with high statistical certainty only for the COG treatment, which showed the steepest increase in log odds over B trials (Treatment:I[Trial − 1]: COG: est. = 0.73, $p < 0.001$, POS: est. = 0.41, $p = 0.07$, ISO: est. = 0.38, $p = 0.07$; electronic supplementary material, table S2). Treatment differences with respect to these increases (slopes) where, however, statistically not supported (electronic supplementary material, table S2). The estimated variance components for the random effects (electronic supplementary material, table S3) indicated large deviances among *Individuals* (s.d. = 0.76), among *Selection lines* (s.d. = 0.35), and among *Sites* (s.d. = 0.20). These deviances were within the range of the absolute values of the (statistically not supported) treatment contrasts ( | est. | ≤ 0.87; electronic supplementary material, table S2).

The linear mixed model for *Latency* to cross the fence also did not statistically support treatment differences in A (POS–COG: $p = 0.80$, ISO–COG: $p = 0.84$, ISO–POS: $p = 0.96$; electronic supplementary material, table S4) or B trials (POS–COG: $p = 0.19$, ISO–COG: $p = 0.67$, ISO–POS: $p = 0.37$; electronic supplementary material, table S4). The estimation of the variance components indicated deviances among *Individuals* (s.d. = 0.59) and among *Residuals* (s.d. = 1.07; electronic supplementary material, table S5) that were larger than the absolute values of the (statistically not supported) treatment contrasts ( | est. | ≤ 0.4; electronic supplementary material, table S4).

## 3.2. Problem-solving test

In the PST, we recorded the occurrence of interactions with the lid as variables *Touched* and *Opened* (figure 5) and the corresponding latencies as *LatencyT* and *LatencyO* (figure 6). In all these behavioural measures, no consistent patterns with respect to the treatment groups were apparent, but the goats differed largely between selection lines and sites. The average number of touches of the treatment groups ranged from 0 to 2.5 for dwarf goats and from 3.0 to 4.4 for dairy goats depending on selection lines and site (figure 5a). In DU, dwarf goats from the ISO group never touched the container. For both selection lines, the average number of touches for each treatment group was consistently higher in ET than in DU. At both sites, none of the dwarf goats opened the container

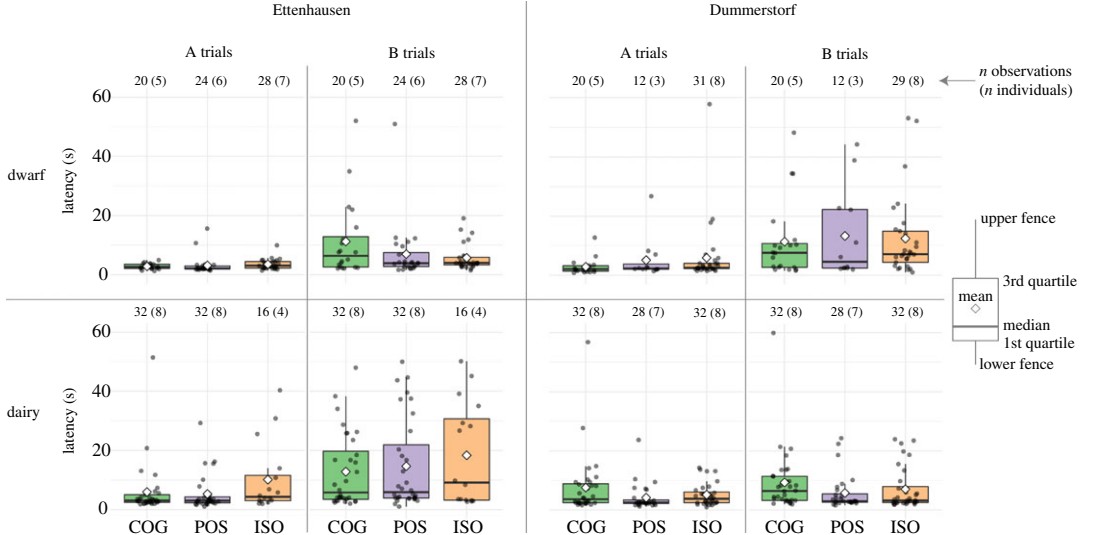

**Figure 4.** Latency to cross the fence for dwarf (top panels) and dairy goats (bottom panels) of the COG, POS and ISO treatment groups in Ettenhausen (four left panels) and Dummerstorf (four right panels). Distribution of individual latencies (jittered points) are summarized as boxplots. As in the statistical analysis, goats that showed an incorrect response in A3 and A4 trials are excluded from this figure.

(data not shown in figures). The average number of trials for each treatment group in which a dairy goat opened the container was always smaller than 0.9 in DU, whereas it ranged from 3.2 to 3.9 in ET.

Consistent for all treatment groups, in ET the average *LatencyT* was around 5 s in dairy goats and 10 s or more in dwarf goats (figure 6*a*). Because dwarf goats in DU almost never touched the lid, corresponding latencies could be measured on only a few occasions. Only the average *LatencyT* for the POS dairy goats in DU was similar to the *LatencyT* of the corresponding treatment group in dairy goats in ET (figure 6*a*). In ET, shortly after the dairy goats touched the lid, most of them also opened it (*LatencyO*, figure 6*b*). In DU, the few dairy goats that opened the lid had on average a similar *LatencyO* as the dairy goats in ET.

In the GLMM for *Touched*, no effect of treatment on the probability of the animals to touch the container was detected (POS–COG: $p = 0.37$, ISO–COG: $p = 0.23$, ISO–POS: $p = 0.71$; electronic supplementary material, table S6). The probability for *Touched* decreased over trials in all groups, but with varying statistical certainty (Treatment:I[Trial − 1]: COG: est. = −0.28, $p = 0.12$, POS: est. = −0.70, $p < 0.001$, ISO: est. = −0.63, $p = 0.003$; electronic supplementary material, table S6). Estimated variance components indicated deviances among *Selection lines* (s.d. = 1.51), among *Sites* (s.d. = 0.92) and among *Individuals* (s.d. = 1.10; electronic supplementary material, table S7) that were similar to or larger than the absolute values of the (statistically not supported) treatment contrasts ( | est. | ≤ 0.98; electronic supplementary material, table S6). Similarly, in the GLMM for *Opened*, no effect of treatment on the probability of the goats to open the container was detected (POS–COG: $p = 0.65$, ISO–COG: $p = 0.58$, ISO–POS: $p = 0.33$; electronic supplementary material, table S8). The estimated variance components indicated deviances among *Individuals* (s.d. = 3.97) and among *Sites* (s.d. = 4.28; electronic supplementary material, table S9) that were larger than the absolute values of the (statistically not supported) treatment contrast ( | est. | ≤ 2.35; electronic supplementary material, table S8).

The linear mixed model did not statistically detect differences in *LatencyT* between the COG, POS and ISO treatments (POS–COG: $p = 0.18$, ISO–COG: $p = 0.93$, ISO–POS: $p = 0.21$; electronic supplementary material, table S10). The model estimated small increases in *LatencyT* of 7–15% (0.1–0.2 log$_2$ units) with every additional trial for all treatments (Treatment:I[Trial − 1]: COG: est. = 0.12, $p = 0.05$, POS: est. = 0.21, $p = 0.001$, ISO: est. = 0.16, $p = 0.01$; electronic supplementary material, table S10). The estimation of variance components suggested deviances among *Selection lines* (s.d. = 0.74), among *Individuals* (s.d. = 0.48) and among *Residuals* (s.d. = 0.75; electronic supplementary material, table S11) that were larger than the absolute values of the (statistically not supported) treatment contrasts ( | est. | ≤ 0.34; electronic supplementary material, table S10). For *LatencyO* in dairy goats, the linear mixed model suggested longer latency to open the lid in the POS versus the ISO treatment (POS–COG: $p = 0.32$, ISO–COG: $p = 0.22$, ISO–POS: $p = 0.03$; electronic supplementary material, table S12). With increasing trial number, *LatencyO* was estimated to decrease by 15–41% (0.2–0.5 log$_2$ units) in all treatments (Treatment:I[Trial − 1]: COG: est. = −

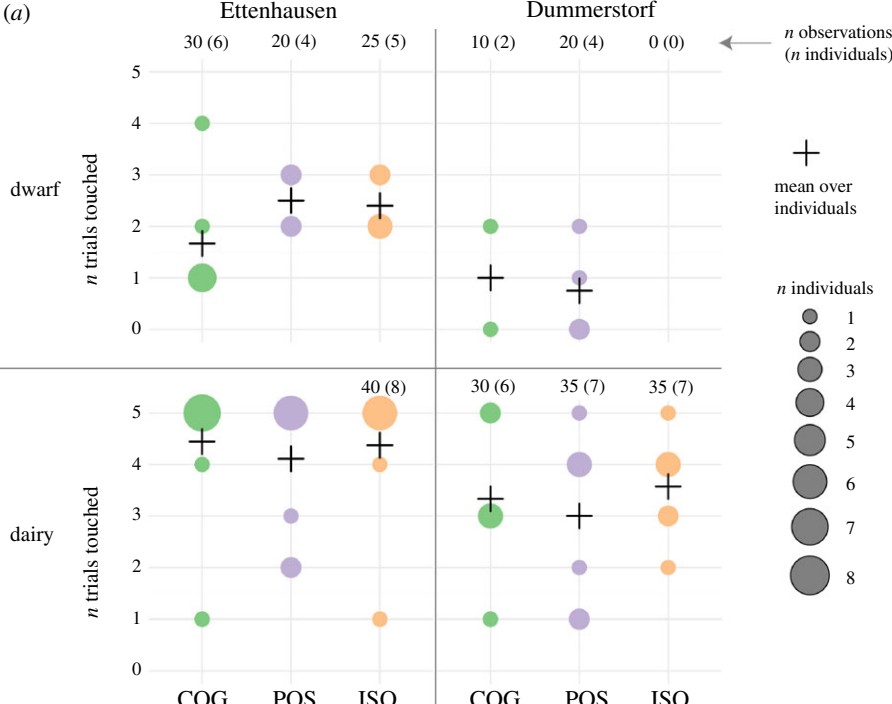

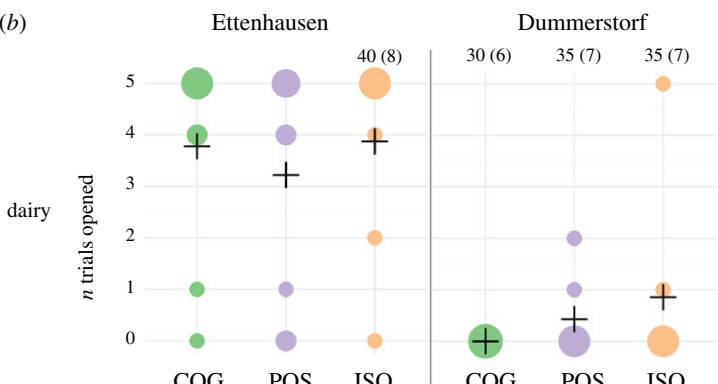

**Figure 5.** Number of trials in which dwarf (top panels) and dairy goats (bottom panels) of the COG, POS and ISO treatment groups (*a*) touched and (*b*) opened the container in Ettenhausen (left) and Dummerstorf (right). The cross indicates the mean, and the size of circles indicates the number of animals that chose the correct side. If not indicated otherwise, observations (*n* = 45) and animals (*n* = 9) were complete.

0.35, $p < 0.001$, POS: est. = −0.51, $p < 0.001$, ISO: est. = −0.24, $p = 0.02$; electronic supplementary material, table S12). Estimated variance components indicated deviances among *Sites* (s.d. = 0.55) and among *Residuals* (s.d. = 0.84; electronic supplementary material, table S13) of similar magnitude as the absolute values of the estimates for the treatment contrasts (|est.| = 0.84; electronic supplementary material, table S12).

## 4. Discussion

We investigated whether participation in cognitive tests (here: visual discrimination and reversal learning as well as a cognitive test battery consisting of object-choice tests) over a period of four to five months affects the performance of goats in subsequent conceptually different cognitive tests, namely a spatial ABT and an instrumental PST. By comparing three treatment groups (COG, POS, ISO), we aimed to disentangle potential effects of the preceding cognitive test exposure (COG–POS) from effects of positive human–animal interactions (POS–ISO). We did not find notable differences between our treatment groups in terms of their behavioural flexibility in the ABT or their performance in the PST. Our results are thus in line with previous research suggesting that cognitive test exposure does not

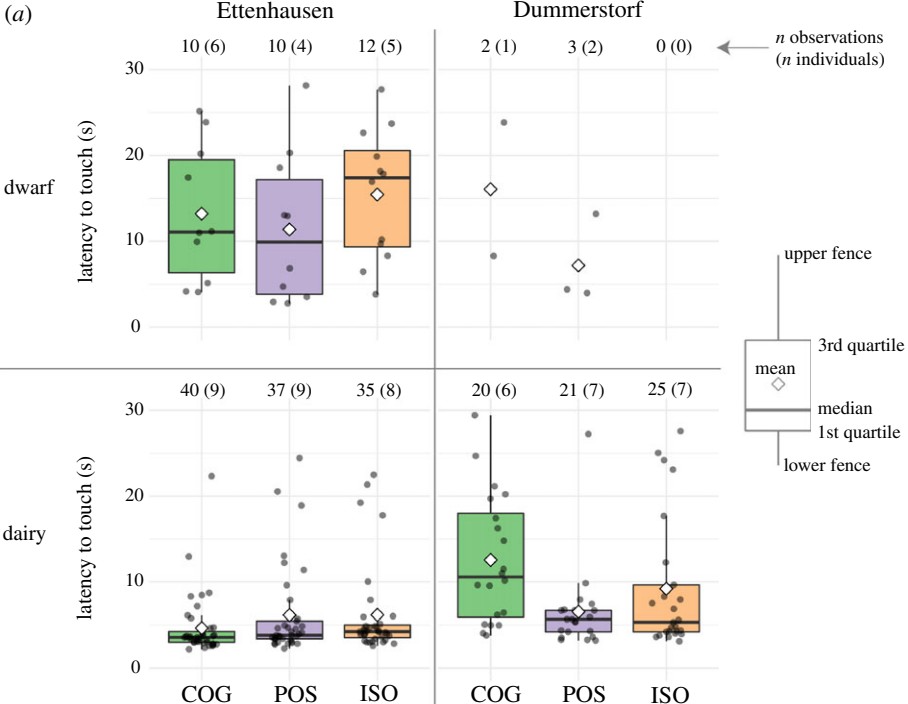

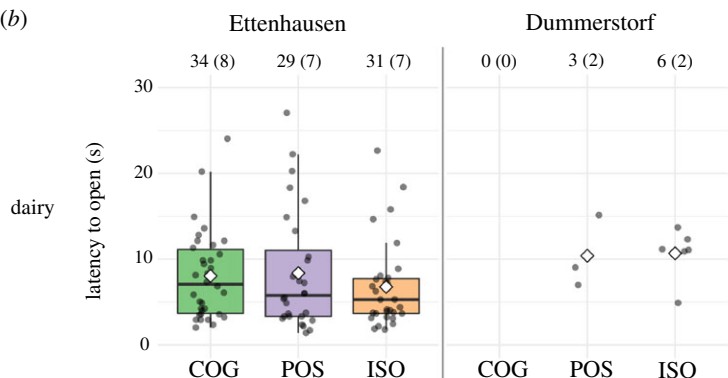

**Figure 6.** (a) *LatencyT* to touch the container in dwarf (top panels) and dairy goats (bottom panels) and (b) *LatencyO* to open the container in dairy goats in Ettenhausen (left) and Dummerstorf (right) from the COG, POS and ISO treatment groups. Distribution of individual latencies (jittered points) are summarized as boxplots.

substantially alter performance in subsequent conceptually different tests (e.g. [6,7,47]). By contrast to the treatments, individuals, selection lines, and sites accounted for large proportions of the variation in our data (see electronic supplementary material, tables S3, S5, S7, S9, S11, S13).

We hypothesized that cognitive training via object-choice tests does improve the performance in a subsequent spatial detour test (the ABT) and, therefore, expected the COG treatment group to be more behaviourally flexible in the ABT than the other treatment groups, i.e. to detour the fence more often correctly and faster. However, we did not find support for this hypothesis. All treatment groups were similarly affected by the change in placement of the spatial barrier and had longer latencies and fewer correct responses in B1 than in A4 trials, indicating a spatial perseveration error. The lack of a treatment effect between COG and POS suggests that the cognitive test experience *per se* did not affected the goats' ability to better inhibit their initially learned response in the B trials over time. The lack of a treatment effect between POS and ISO furthermore suggests that the additional positive human–animal interaction in the COG and POS groups had not substantially affected the goats' behaviour in the spatial ABT. As discussed in Langbein *et al.* [48], maybe the normal daily handling during husbandry procedures had already altered the behaviour of the goats of all treatment groups, interfering with the effect of the additional human contact during the application of the treatments.

In the PST, we hypothesized that the COG treatment enhances goats' problem-solving ability and thus that these goats would touch and open a covered container more often and faster than POS goats. Our results do not support this hypothesis as no COG versus POS differences in any of the variables in the PST were found. This finding is in contrast to studies on dogs which showed that high levels of training generally improved the dogs' problem-solving abilities and their probability to interact with novel objects [8–10]. However, trained dogs often experience different management conditions to non-trained dogs, and they may show improved performance due to other reasons than cognitive test experience *per se*. In addition, we expected that POS goats also perform better than goats from the ISO treatment. However, POS dairy goats showed an even longer latency to open the lid than ISO dairy goats. In our study, the probability to touch the container decreased over trials, suggesting that the goats may have lost interest in the container with increasing trial number. Again, this development occurred similarly in all treatment groups. The motivation to explore and learn is a strong predictor of problem-solving success [49], but it may decrease with increasing exposure to the novel item [50,51], given the problem is not kept challenging enough by regular modification [52]. In sum, these results suggest that cognitive test exposure did not substantially affect the outcomes of the ABT and the PST, even though exposure to the cognitive testing was longer than in other studies (e.g. [6,7]).

To increase the heterogeneity of our sample and thus the external validity of our findings, we tested two selection lines of goats (dairy goats and dwarf goats) at two research sites [32–34]. In the current study, we found a high variance of *Site* and *Selection line* which highlights the importance of effects of location as well as phenotypic variation on cognitive test results. We found site differences within dwarf goats although subjects from both sites originated from mainly the same population, indicating that factors other than genetics should be considered as well. Because we used almost the same experimental set-up and the same human experimenters at both sites, this variation might for example be caused by different previous experiences with animal care staff at the two research sites [53,54], the size of the farms where the goats were bred [55], the rearing history, i.e. human- or dam-reared [56] or the isolation and handling at a young age [57].

Possible explanations for the phenotypic variations may be genetic and/or environmental factors. Van Horik & Madden [7] found that inherent motivational traits, such as the motivation to approach an apparatus or the persistency to solve a test, best predict the success in a PST. These traits have often been linked to personality traits. Dogs described as active, playful and bold were found to be more likely to learn complex behaviours and perform well in situations requiring persistence [58]. Similarly, bold and exploratory animals were found to be particularly likely to be innovative [59–61] and to overcome novel challenges faster compared with less bold animals [62]. Learning speed in reversal tests was also found to be correlated with individual variation in exploration, but this relationship differed between tests and age [63]. Interestingly, personality seems to affect learning performances especially if individuals are in a state of stress [64]. Thus, the differences we found in the PST in dwarf goats compared with dairy goats may also be explained by other factors such as different levels of stress, which were found to be linked to the motivation to explore [65].

Additionally, fear towards the experimenter may have affected performance in the PST [14] by reducing the motivation of stressed individuals to approach the container positioned next to the experimenter. Although both selection lines were handled in a similar manner, dairy goats might have been more inclined to approach humans during training and test sessions compared with dwarf goats. Research in chickens and sheep suggests that selection for high productivity has reduced stress reactivity towards humans [66,67]. If we apply this assumption to our study, it seems that the selection for high productivity in dairy goats may have decreased fear towards humans and as a result also increased the goats' inclination to approach and manipulate the container positioned next to the experimenter. As is common for the dairy industry, the dairy goats used in this study had been separated from their dam right after birth. By contrast, dwarf goats had been allowed to stay with their mothers for six weeks. Early separation from the dam and rearing by humans has been shown to increase tameness scores in goats [68].

When comparing the variances in performances of the two tests applied in this study, we found that the ABT showed less variation in factors such as *Site* and *Selection line* compared with the PST (see electronic supplementary material, tables S3, S5, S7, S9, S11, S13). By contrast to the PST, in the ABT both, dwarf and dairy goats, were able to solve the problem (i.e. to detour the fence). This finding may indicate that the standardized ABT set-up is better suited for ungulate species [25,31] than the PST set-up that was used in our study. Nevertheless, both tests seem to assess biologically relevant skills for goats as a species. Goats show highly selective feeding behaviour adapted to seasonal

changes in plant abundance [69]. Hence, they need to have the ability to manipulate specific plants and to overcome spatial barriers to access feed sources that are out of reach. Indeed, domestic goats were shown to possess good spatial learning abilities in a maze learning paradigm [70] and were found to be capable of opening containers in other studies [27,28,48]. The manipulation test in this study might have induced different levels of neophobia between selection lines and thus probably would have required higher levels of habituation to the set-up for dwarf goats. Even though the suitability of different test paradigms cannot be conclusively answered with our study, our findings suggest that caution must be taken when making general claims about cognitive capacities of a species from studies using a single selection line or a narrow range of phenotypes.

# 5. Conclusion

Our results suggest that cognitive testing *per se* and the positive association with the human during testing do not notably affect the performance in subsequent conceptually different cognitive tests in goats. Furthermore, we found that variability among selection lines and research sites can be considerable in spatial detour tests and instrumental PSTs. Therefore, we suggest that further cognitive research should conduct multi-site studies and consider a broad range of phenotypes of a species.

Ethics. All animal care and experimental procedures were performed in accordance with all relevant Swiss legislative and regulatory requirements as well as the German welfare requirements for farm animals and the ASAB/ABS Guidelines for the Use of Animals in Research [37]. All procedures involving animal handling and treatment were approved by the Cantonal Veterinary Office, Thurgau, Switzerland (Approval no. TG04/17–29343) and the Committee for Animal Use and Care of the Ministry of Agriculture, Environment and Consumer Protection of the federal state of Mecklenburg-Vorpommern, Germany (Approval no. 7221.3-1.1-062/17).

Data accessibility. Data from tests and animals as well as the R code used to perform the analyses described are provided as electronic supplementary material [71].

Authors' contributions. K.R. and C.N. collected the data. M.S. and K.R. analysed the data. K.R. drafted the manuscript and wrote the main parts of the manuscript. M.S. provided statistical support and made all corresponding statistical plots. C.N., N.K. and J.L. conceptualized the study and provided supervision. All authors interpreted the data and provided critical feedback on the manuscript.

Competing interests. The authors declare no competing interests.

Funding. This work was supported by an SNF grant (no. 310030E-170537) to N.K. and a DFG grant (no. LA 1187/6-1) to J.L.

Acknowledgements. We thank the staff of the Agroscope Research Station in Ettenhausen, Switzerland, and of the Research Institute for the Biology of Farm Animals in Dummerstorf, Germany, for assisting in this experiment and for caring for the animals. We additionally want to thank Dieter Sehland and Jonas Barkhau for coding our videos.

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
