## [Peer Review File · Royal Society Open Science]

Review History

RSOS-210656.R0 (Original submission)

Review form: Reviewer 1

Is the manuscript scientifically sound in its present form?

Yes

Are the interpretations and conclusions justified by the results?

Yes

Is the language acceptable?

Yes

Do you have any ethical concerns with this paper?

No

Have you any concerns about statistical analyses in this paper?

Yes

Recommendation?

Accept with minor revision (please list in comments)

Comments to the Author(s)

This manuscript aims at investigating the effect of previous exposure to training, including isolation from other individual, food rewards provided by humans and cognitive tests, on performance of 2 goat lines (dwarf and dairy) during 2 standard cognitive tasks (detour and instrumental problem-solving test). The results do not show any clear treatment effects, but point towards selection line differences for instance. These results provide valuable knowledge on the impact (or absence of impact) of former training in farm animals used in cognitive research. I have a suggestion for further tests and comments on the design and statistics.

As a general comment, if the authors are interested in differences between selection lines, which have been bred based on various characteristics for several generations, they should maybe consider adding the selection line as an addition fixed effect in their models. This would allow a more direct comparison of performances between lines, which would be very interesting.

L210-214. Were ISO goats exposed to humans during the isolation (was an experimenter present)?

L326-338. The number of animals excluded is unusually high. Aren't the results of some of these excluded animals relevant (e.g. number of goats that didn't perform the task within 60s, or that didn't perform correctly in A3 and A4)?

L359. If I understand correctly, your model doesn't include the main effects alone (Type and Treatment) and instead only includes their interaction? Normally, interactions should not be included in a model without the corresponding main effects.

Review form: Reviewer 2

Is the manuscript scientifically sound in its present form?

Yes

Are the interpretations and conclusions justified by the results?

No

Is the language acceptable?

Yes

Do you have any ethical concerns with this paper?

No

Have you any concerns about statistical analyses in this paper?

Yes

Recommendation?

Accept with minor revision (please list in comments)

Comments to the Author(s)

Please see the attached file (Appendix A).

Decision letter (RSOS-210656.R0)

Dear Mrs Rosenberger,

On behalf of the Editors, we are pleased to inform you that your Manuscript RSOS-210656 "Performance of goats in a detour and a problem-solving test following long-term cognitive test exposure." has been accepted for publication in Royal Society Open Science subject to minor revision in accordance with the referees' reports. Please find the referees' comments along with any feedback from the Editors below my signature.

Please submit your revised manuscript and required files (see below) no later than 7 days from today's (ie 16-Aug-2021) date. Note: the ScholarOne system will 'lock' if submission of the revision is attempted 7 or more days after the deadline. If you do not think you will be able to meet this deadline please contact the editorial office immediately.

on behalf of Dr Rosalind Arden (Associate Editor) and Kevin Padian (Subject Editor)
openscience@royalsociety.org

Associate Editor Comments to Author (Dr Rosalind Arden):

Thank you for submitting your MS to RSOS for review. It took some time for us to identify relevant expert with suitable expertise so you have been patient - thank you for that.

Both Reviewers concur with us that this is a good piece of work which we would like to accept.

Both reviewers make some useful suggestions to strengthen the ms. We would like those to be taken into account in your revision. Most of these comments are both substantive yet easily addressed. For example see Rev 1 L359 comment. The care you have taken in your work is noted by the Reviewers who have done a great job of reading and responding to the ms in detail.

Reviewer comments to Author:

Reviewer: 1

Comments to the Author(s)

This manuscript aims at investigating the effect of previous exposure to training, including isolation from other individual, food rewards provided by humans and cognitive tests, on performance of 2 goat lines (dwarf and dairy) during 2 standard cognitive tasks (detour and instrumental problem-solving test). The results do not show any clear treatment effects, but point towards selection line differences for instance. These results provide valuable knowledge on the impact (or absence of impact) of former training in farm animals used in cognitive research. I have a suggestion for further tests and comments on the design and statistics.

As a general comment, if the authors are interested in differences between selection lines, which have been bred based on various characteristics for several generations, they should maybe consider adding the selection line as an addition fixed effect in their models. This would allow a more direct comparison of performances between lines, which would be very interesting.

L210-214. Were ISO goats exposed to humans during the isolation (was an experimenter present)?

L326-338. The number of animals excluded is unusually high. Aren't the results of some of these excluded animals relevant (e.g. number of goats that didn't perform the task within 60s, or that didn't perform correctly in A3 and A4)?

L359. If I understand correctly, your model doesn't include the main effects alone (Type and Treatment) and instead only includes their interaction? Normally, interactions should not be included in a model without the corresponding main effects.

Reviewer: 2

Comments to the Author(s)

Please see the attached file "RSOS-210656 review.pdf".

===PREPARING YOUR MANUSCRIPT===

While not essential, it will speed up the preparation of your manuscript proof if you format your references/bibliography in Vancouver style (please see

<https://royalsociety.org/journals/authors/author-guidelines/#formatting>). You should include DOIs for as many of the references as possible.

===PREPARING YOUR REVISION IN SCHOLARONE===

<https://royalsociety.org/journals/authors/author-guidelines/#data>. You should ensure that you cite the dataset in your reference list. If you have deposited data etc in the Dryad repository,

please only include the 'For publication' link at this stage. You should remove the 'For review' link.

Author's Response to Decision Letter for (RSOS-210656.R0)

See Appendix B.

Decision letter (RSOS-210656.R1)

Dear Mrs Rosenberger,

I am pleased to inform you that your manuscript entitled "Performance of goats in a detour and a problem-solving test following long-term cognitive test exposure." is now accepted for publication in Royal Society Open Science.

You can expect to receive a proof of your article in the near future. Please contact the editorial office (openscience@royalsociety.org) and the production office (openscience_proofs@royalsociety.org) to let us know if you are likely to be away from e-mail contact -- if you are going to be away, please nominate a co-author (if available) to manage the proofing process, and ensure they are copied into your email to the journal. Due to rapid

publication and an extremely tight schedule, if comments are not received, your paper may experience a delay in publication.

on behalf of Dr Rosalind Arden (Associate Editor) and Kevin Padian (Subject Editor)
openscience@royalsociety.org

Appendix A

MS RSOS-210656 by Rosenberger et al.

Performance of goats in a detour and a problem-solving test following long-term cognitive test exposure.

This is a valuable manuscript. It reports a well-designed and well-conducted experiment, with a substantial sample size, to investigate a question that is important but might be thought a little dull – does exposing the same individuals to repeated cognitive testing affect their performance on those tests, even if the tests concerned are apparently unrelated? As the authors document, it is common practice in animal cognition to test the same individuals repeatedly, and the longer-lived and more exotic and expensive the species concerned, the more likely this is to happen. Given that longer-lived, exotic and expensive species are often those of most scientific and public interest, the question is an acute one.

The strengths of the paper lie in the facts that (a) the authors carefully considered what factors might contribute to an apparent improvement in performance with repeated testing, so they included control groups that experienced some of the manipulations involved in testing but not the tests themselves; and (b) they included some ecologically valid variations between groups – two different cognitive tests, different strains, and different testing sites – of the sort that inevitably arise when we compare experiments by different research groups.

As it turned out, the authors found that previous testing experience did not have any effect on performance in their final tests, whereas both strain and test site did, at least on one of their tests (PST, a problem-solving test involving removing a lid from a familiar food container); there was also great individual variation. Again, that might be thought a dull result. On the contrary, it's very important – and a huge relief to those of us who have spent much time testing the same few animals in different experiments.

In addition to the merits of the experiments and the results, the Introduction provides a valuable review of the existing literature on the impacts on animals of cognitive testing, and the procedures associated with it.

The paper is clearly and professionally written. I only have two concerns about the information provided, and one is trivial.

1. In the PST, I could not find it stated whether the food container was fixed, to the ground or the wall of the enclosure. This makes quite a difference to the possible manipulations the goats could have used to dislodge the lid.
2. The only suggestion of an effect of previous cognitive training was in the A-not-B detour test (ABT), where goats with prior cognitive experience (COG group) showed a significant improvement in performance across the four trials in the B condition, whereas other goats did not (although the trend was in the same direction), and this is discussed as indicated that perhaps the COG groups learned faster (lines 398-400, 487-489). Comparing statistical significance levels in this way is an error: difference of significance does not indicate significance of difference. The appropriate procedure is to carry out a test of the interaction between trial number and experience group. From the formula in lines 359-360, I don't think such an interaction was included in the model (though I

am not an expert in R model formulae). If the authors want to examine learning rates, they need to include interactions in their model. They also need to show us the trends in proportions of trials correct as a function of trial number – it would be possible for the trend to be identical in the three experience groups, and yet for there to be a difference in significance, because of differences in individual variation.

As an aside, the need for goats to manipulate specific plants (lines 562-563) is interesting, and put me in mind of Byrne's studies of gorilla food technology (e.g. Byrne & Byrne 1993, *Amer J Primatology*). It's probably too remote a connection to bring into the paper, however.

As regards presentation, I only have one substantive suggestion. At line 135, the authors don't quite bring out that using two different research sites is a strength of their design (though they recognize this at lines 146-148). I suggest changing "the same two" in line 135 to "two different" to make the point.

The English of the paper is excellent, clear, unfussy and almost entirely correct. In just a few places it isn't quite idiomatic. I noticed the following places where the authors could make the text sound more natural. I am sure there are other similar instances, but these points are so minor that further copy-editing is quite unnecessary.

Line

- 29 "conditioned" is odd. More natural would be "used"
- 53 "the participation" – drop "the"
- 61 "The frequent" – drop "The"
- 71 "hardly" isn't quite right. You could say "hardly at all" but that would be fussy. The best word in context is probably "barely".
- 77 change "have not been" to "was"
- 150 insert "a" before "pet"
- 153 change "does likely not exceed" to "probably does not exceed"
- 197 delete "à"
- 204 Are you American or European? In this sort of context, Americans use singular verbs after "group" (because there's just one group); European English speakers use plural verbs (because there are multiple individuals in the group). Take your choice!
- 332 Change "per definition" to "by definition"
- 545 Change "chicken" to "chickens"
- 549 Change "As it is common" to "As is common"

Appendix B

Performance of goats in a detour and a problem-solving test following long-term cognitive test exposure

Response to reviewer

Dear Editor,

Thank you for the thoughtful and constructive comments from you and the two reviewers on our manuscript 'Performance of goats in a detour and a problem-solving test following long-term cognitive test exposure.' As requested, we have revised the manuscript, taking careful account of all comments made by yourself and the two reviewers. Together with this revision note, we have resubmitted a revised version of the manuscript with all changes highlighted.

We hope that the present version of the manuscript has improved significantly and that you might consider this manuscript now for publication in *Royal Society Open Science*.

The material in this manuscript has not been published elsewhere and is not submitted for publication elsewhere. All authors have seen the final manuscript and we all take responsibility for its contents.

Sincerely,

Katrina Rosenberger, Michael Simmler, Jan Langbein, Nina Keil and Christian Nawroth,

Performance of goats in a detour and a problem-solving test following long-term cognitive test exposure

Associate Editor Comments to Author (Dr Rosalind Arden):

Thank you for submitting your MS to RSOS for review. It took some time for us to identify relevant expert with suitable expertise so you have been patient - thank you for that.

Both Reviewers concur with us that this is a good piece of work which we would like to accept.

Both reviewers make some useful suggestions to strengthen the ms. We would like those to be taken into account in your revision. Most of these comments are both substantive yet easily addressed. For example see Rev 1 L359 comment. The care you have taken in your work is noted by the Reviewers who have done a great job of reading and responding to the ms in detail.

Authors' response: Thank you for the positive feedback!

Performance of goats in a detour and a problem-solving test following long-term cognitive test exposure

Reviewer Comments:

Reviewer 1

This manuscript aims at investigating the effect of previous exposure to training, including isolation from other individual, food rewards provided by humans and cognitive tests, on performance of 2 goat lines (dwarf and dairy) during 2 standard cognitive tasks (detour and instrumental problem-solving test). The results do not show any clear treatment effects, but point towards selection line differences for instance. These results provide valuable knowledge on the impact (or absence of impact) of former training in farm animals used in cognitive research. I have a suggestion for further tests and comments on the design and statistics.

Authors' response: Thank you for the positive feedback!

As a general comment, if the authors are interested in differences between selection lines, which have been bred based on various characteristics for several generations, they should maybe consider adding the selection line as an additional fixed effect in their models. This would allow a more direct comparison of performances between lines, which would be very interesting.

Authors' response: More than one selection line were included in our study to increase heterogeneity among goats in order to get a more robust inference towards our effects of interest, which are the treatment effects (see description of goals in introduction). During the design of the study, we did not anticipate to compare the performance of both selection lines and think that adding this term ad hoc as additional fixed effect would be a deviation from the initial study rationale. We think our models best reflect our initial intention.

L210-214. Were ISO goats exposed to humans during the isolation (was an experimenter present)?

Authors' response: We rephrased the text accordingly.

Line 210-215: "Individuals allocated to the ISO treatment neither participated in cognitive tests nor did they receive rewards by the experimenter. However, they were isolated over a similar period as the COG and the POS group in the same arena (= median time taken by COG group to finish all trials in the previous test session) and with the experimenter present behind the crate, as was the case for the COG and POS treatments."

L326-338. The number of animals excluded is unusually high. Aren't the results of some of these excluded animals relevant (e.g. number of goats that didn't perform the task within 60s, or that didn't perform correctly in A3 and A4)?

Authors' response: We agree that drop-out rate and numbers of excluded animals can be interpreted as a considerable amount. Lines 327-339 describe the reasoning behind the large number of animals we have excluded for both tasks. The additional exclusion of animals which did not perform correctly in trials A3 and A4 is motivated by the nature

Performance of goats in a detour and a problem-solving test following long-term cognitive test exposure

of the test, as the A-not-B test assumes that the animals learned to correctly perform the task over the A trials. We have outlined this in the text in Lines 330-342.

L359. If I understand correctly, your model doesn't include the main effects alone (Type and Treatment) and instead only includes their interaction? Normally, interactions should not be included in a model without the corresponding main effects.

Authors' response: We considered a treatment effect over A and B type trials together as not meaningful (i.e. it is likely that different decision-making processes are at play for the different trial types) and therefore analysed this effect separately for A and B type trials. Therefore, the fixed effects were specified as interaction with the categorical variable 'type'.

Reviewer 2

This is a valuable manuscript. It reports a well-designed and well-conducted experiment, with a substantial sample size, to investigate a question that is important but might be thought a little dull – does exposing the same individuals to repeated cognitive testing affect their performance on those tests, even if the tests concerned are apparently unrelated? As the authors document, it is common practice in animal cognition to test the same individuals repeatedly, and the longer-lived and more exotic and expensive the species concerned, the more likely this is to happen. Given that longer-lived, exotic and expensive species are often those of most scientific and public interest, the question is an acute one.

The strengths of the paper lie in the facts that (a) the authors carefully considered what factors might contribute to an apparent improvement in performance with repeated testing, so they included control groups that experienced some of the manipulations involved in testing but not the tests themselves; and (b) they included some ecologically valid variations between groups – two different cognitive tests, different strains, and different testing sites – of the sort that inevitably arise when we compare experiments by different research groups.

As it turned out, the authors found that previous testing experience did not have any effect on performance in their final tests, whereas both strain and test site did, at least on one of their tests (PST, a problem-solving test involving removing a lid from a familiar food container); there was also great individual variation. Again, that might be thought a dull result. On the contrary, it's very important – and a huge relief to those of us who have spent much time testing the same few animals in different experiments.

In addition to the merits of the experiments and the results, the Introduction provides a valuable review of the existing literature on the impacts on animals of cognitive testing, and the procedures associated with it.

Authors' response: Thank you for the positive feedback!

The paper is clearly and professionally written. I only have two concerns about the information provided, and one is trivial.

Performance of goats in a detour and a problem-solving test following long-term cognitive test exposure

1. In the PST, I could not find it stated whether the food container was fixed, to the ground or the wall of the enclosure. This makes quite a difference to the possible manipulations the goats could have used to dislodge the lid.

Authors' response: We have edited the text accordingly.

Lines 277-278: The PST test in our study is an instrumental manipulation test that requires the animal to open a familiar unfixed and freestanding food container covered with a lid novel to the animal.

2. The only suggestion of an effect of previous cognitive training was in the A-not-B detour test (ABT), where goats with prior cognitive experience (COG group) showed a significant improvement in performance across the four trials in the B condition, whereas other goats did not (although the trend was in the same direction), and this is discussed as indicated that perhaps the COG groups learned faster (lines 398-400, 487-489). Comparing statistical significance levels in this way is an error: difference of significance does not indicate significance of difference. The appropriate procedure is to carry out a test of the interaction between trial number and experience group. From the formula in lines 359-360, I don't think such an interaction was included in the model (though I am not an expert in R model formulae). If the authors want to examine learning rates, they need to include interactions in their model. They also need to show us the trends in proportions of trials correct as a function of trial number – it would be possible for the trend to be identical in the three experience groups, and yet for there to be a difference in significance, because of differences in individual variation.

Authors' response: We agree with the reviewer that difference of significance does not indicate significance of difference. The interaction with trial number is already in the model. As treatment is a categorical variable the correct test would be the treatment contrast below, which tests for significant difference in the slope over the b trials. The estimate is negative as the slope was steeper for COG versus POS. But, this apparent difference is statistically poorly supported ($p = 0.312$, see below). Therefore, we adjusted the manuscript as follows: We now report the interaction contrasts in the SI (see ESM Table S2). In the results section, we additionally report that there are no statistically supported differences between the slopes of the different treatments (see Lines 407-409). Finally, we edited text in the abstract (Lines 35-37) and discussion (Lines 481-483), and removed text that referred to the improvement in the slopes from the discussion (Lines 494-496).

```
> summary(treatment_contrasts, test = adjusted('none'))

      Simultaneous Tests for General Linear Hypotheses

Fit: glmer(formula = Accuracy ~ 0 + Type:Treatment + Type:Treatment:I(Trial -
1) + (1 | SelectionLine) + (1 | Site/Pen/Individual), data = ABT_data,
family = "binomial", control = glmerControl(optimizer = "bobyqa"))

Linear Hypotheses:

                                Estimate Std. Error z
value Pr(>|z|)
Typeb:TreatmentPOS:I(Trial - 1) - Typeb:TreatmentCOG:I(Trial - 1) == 0  -0.3180      0.3143  -
1.012      0.312
(Adjusted p values reported -- none method)
```

Performance of goats in a detour and a problem-solving test following long-term cognitive test exposure

As an aside, the need for goats to manipulate specific plants (lines 562-563) is interesting, and put me in mind of Byrne's studies of gorilla food technology (e.g. Byrne & Byrne 1993, Amer J Primatology). It's probably too remote a connection to bring into the paper, however.

Authors' response: Thank you for the hint towards this interesting paper. However, we agree with the reviewer that it is too remote a connection to bring it into the paper.

As regards presentation, I only have one substantive suggestion. At line 135, the authors don't quite bring out that using two different research sites is a strength of their design (though they recognize this at lines 146-148). I suggest changing "the same two" in line 135 to "two different" to make the point.

Authors' response: Thank you for highlighting this. We have now edited the text according to the reviewer's suggestion (see Line 134).

The English of the paper is excellent, clear, unfussy and almost entirely correct. In just a few places it isn't quite idiomatic. I noticed the following places where the authors could make the text sound more natural. I am sure there are other similar instances, but these points are so minor that further copy-editing is quite unnecessary.

Line

29 "conditioned" is odd. More natural would be "used"

53 "the participation" – drop "the"

61 "The frequent" – drop "The"

71 "hardly" isn't quite right. You could say "hardly at all" but that would be fussy. The best word in context is probably "barely".

77 change "have not been" to "was"

150 insert "a" before "pet"

153 change "does likely not exceed" to "probably does not exceed"

197 delete "à"

204 Are you American or European? In this sort of context, Americans use singular verbs after "group" (because there's just one group); European English speakers use plural verbs (because there are multiple individuals in the group). Take your choice!

332 Change "per definition" to "by definition"

545 Change "chicken" to "chickens"

549 Change "As it is common" to "As is common"

Authors' response: We have amended all minor corrections. Thanks for flagging those!